# The Effect of Ice on Shoulder Proprioception in Badminton Athletes

Joel Marouvo [1,2,*], Nuno Tavares [1], Gonçalo Dias [3,4,5,6,7] and Maria António Castro [1,2,8]

1   RoboCorp, i2A, Polytechnic Institute of Coimbra, 3045-093 Coimbra, Portugal
2   Centre for Mechanical Engineering, Materials and Processes (CEMMPRE), University of Coimbra, 3030-788 Coimbra, Portugal
3   ESEC-UNICID-ASSERT, Instituto Politécnico de Coimbra, 3030-329 Coimbra, Portugal
4   ROBOCORP, IIA, Instituto Politécnico de Coimbra, 3030-329 Coimbra, Portugal
5   Faculty of Sport Sciences and Physical Education, University of Coimbra, 3040-256 Coimbra, Portugal
6   CIDAF (UID/DTP/04213/2020), Universidade de Coimbra, 3040-248 Coimbra, Portugal
7   Instituto de Telecomunicações, Delegação da Covilhã, 6201-001 Covilhã, Portugal
8   Sector of Physiotherapy, School of Health Sciences, Polytechnic Institute of Leiria, 2411-901 Leiria, Portugal
*   Correspondence: duartemarouvo@gmail.com; Tel.: +351-912-942-487

**Abstract:** This study aims to analyze the influence of the application of cryotherapy on shoulder proprioception in badminton athletes. Thirty federated badminton athletes were included in this study, all of whom belonged to three of the teams currently competing in national competitions (Portugal). Their mean ages were $21.00 \pm 5.60$ years, and their experience in the modality was $8.40 \pm 6.93$ years. They practiced in an average of $2.93 \pm 1.26$ training sessions per week. All of them used their right hand to hold the racket. Each participant's dominant shoulder joint position and force senses were evaluated for four consecutive time points through the isokinetic dynamometer Biodex System 3. The experimental procedure consisted of applying ice for 15 min and the control procedure consisted of no therapeutic intervention. The proprioception outcomes were expressed using the absolute error, relative error, and variable error. All statistical analysis was performed using PASW Statistics 18 software (IBM-SPSS Statistics). There were no statistically significant changes in the joint position and force senses after the intervention, as well as during the subsequent 30 min. We conclude that, after this cryotherapy technique, there is no increased risk of injury associated with a proprioception deficit that prevents athletes from immediately returning to badminton practice.

**Keywords:** cryotherapy; joint position sense; force sense; proprioception

## 1. Introduction

Proprioception is based on the transmission of afferent information from muscles, tendons, joint capsules, ligaments, and the skin to the Central Nervous System (CNS) [1]. For a certain motor action to be performed successfully, the afferent information must be correctly transmitted to the CNS, through the visual, vestibular, and somatosensory sensory systems. The processing of this afferent information, performed at the central level, will later be used to generate an efferent response that allows the proper regulation of muscle tone, ensuring correct patterns of joint stability, coordination, and balance during human movement [2]. Analogously, it is possible to compare the peripheral mechanoreceptors to the hardware of a computer, which provides proprioceptive information for the software (central processing) to integrate and use [3]. Through this process, an individual can compare their real movement with their intended movement, and thus develop their motor learning skills [1], i.e., develop new movement patterns by processing the appropriate proprioceptive information [3].

To carry out a quantitative assessment of proprioception, three components are measured: joint position sensation (JPS), kinesthesia, and muscle tension sensation (MTS).

JPS assesses the precision and accuracy of repositioning a joint at a predetermined angle, kinesthesia measures the ability to perceive movement in the joint, and MTS assesses the ability to perceive and produce a predetermined and previously reproduced submaximal force [1,4,5]. The results from these evaluations are usually the calculation of the absolute error (AE), relative error (RE) and variable error (VE). The AE indicates the difference between the real angle and the target angle, without considering the direction of this difference; the RE transmits the difference between the actual angle and the target angle, considering the direction of this difference, and the VE is obtained through the standard deviation of the different relative error values [1,6].

In order to assess proprioception in the upper and lower limbs, three to five repetitions of each test were performed for each component [1]. However, when it comes to the evaluation of proprioception in the spine, it is recommended that each measurement is repeated five times [1]. There are a series of precautions that must be considered in order to reduce the error associated with these measurements. The amplitude and speed of the evaluated movement must be constant and adjusted to the functional activity that is being evaluated [1]. Musculoskeletal disorders that cause pain, stroke, trauma or fatigue modify the assessed proprioception values, so results in these conditions lack precision [2]. The fact that proprioceptive information is integrated with information from the visual and vestibular systems means that certain care is required during the assessment of proprioception, such as placing a blindfold over the eyes to reduce visual information or using noisy headphones with white noise to limit vestibulocochlear information [1,6].

Some investigations have been carried out to understand the implications of cryotherapy methods on proprioception [7], although there has been no consensus amongst authors. The vast majority of studies use a sample of normal individuals and in the end, generalize the conclusions obtained to a sporting context. These investigations mainly analyze the JPS component and the knee joint, leaving doubt about whether the results found will be similar in the other components of proprioception, or whether they will vary from joint to joint.

Badminton is a racket sport in which there is no contact with the opponent and athletes are required to repeatedly perform jumps, lunges, changes in direction, and quick movements of the upper limbs in different postural positions [8]. In this sport, injuries occur mainly in the ankle, knee, and shoulder [8,9]. These are usually traumatic joint/muscular problems [10–13], or chronic injuries that result from the overload imposed on joints during the continuous repetition of the gestures characteristic of this sport [14,15]. The application of ice is a typical procedure used by physical therapists in these clinical contexts [4,16–19]. However, its use causes a decrease in the speed of the conduction of the nerve impulse in sensory afferent fibers, as well as in the excitability of muscle mechanoreceptors [4,16–18]. This physiological mechanism may alter the athlete's proprioception, leading to an increase in the risk of injury, especially in cases where the cryotherapy method is applied just before sports practice.

The application of different cryotherapy methods is frequently used in the clinical practice of physiotherapy in a sports environment, as its analgesic and anti-inflammatory properties have been associated with a faster recovery from injuries in the neuro-musculoskeletal system [4,16–19]. Cryotherapy serves as a common treatment method for acute sports injuries. Analgesia, inflammation reduction and secondary hypoxic injuries are commonly treated with this method. When an athlete returns to the sport immediately after a serious injury, the benefits of cryotherapy might be questioned, since suppressing pain sensations could lead to more tissue damage. In addition, a study suggests that cryotherapy does not reduce delayed-onset muscle soreness following endurance and strength training [20].

From a physical point of view, cryotherapy's main objective is to transfer thermic energy to the outside of the body. This effect is dependent on several factors, such as the area and duration of the application of the cryotherapy agent, or the depth and cooling technique employed [16]. The most common biological implications are a decrease in temperature and cell metabolism, and an increase in vasoconstriction and local tissue

stiffness. There is also a delay in the nerve impulse conduction speed of the sensorial afferent fibers, as well as a decrease in the excitability of the muscle mechanoreceptors, which will cause alterations both in terms of the perception of stimuli and in the activation of the motor units, compromising the coordination of movement patterns and functional stability [4,16–18]. Most of the studies found to investigate the JPS component present very varied results: in total, ten articles conclude that there are no significant JPS changes after applying ice [6,17,21–28] and nine state the opposite [7,29–34]. The remaining components of proprioception have been less investigated. No article was found to relate cryotherapy with the component of kinesthesia, and only four articles analyzed the effect of a cryotherapy method on MTS [6,35–37]. Of these four studies, only one concluded that the changes resulting from the application of ice significantly influenced the participants' MTS [35].

The main objective of this study is to analyze the influence of the application of a cryotherapy method on shoulder proprioception in badminton athletes.

## 2. Materials and Methods

### 2.1. Participants

This clinical, prospective longitudinal study was carried out at the RoboCorp Laboratory at the Polytechnic Institute of Coimbra after approval from the Ethics Committee of Polytechnic Institute of Coimbra (CEPC 7/2018), based on the revised version of the 2013 Declaration of Helsinki [38,39] Before the assessment, a pilot study was carried out [40] in order to identify the angle of the shoulder flexion at the moment when the badminton racket contacts the shuttlecock during the three shots performed by badminton athletes above the head. Using the data extracted from this preliminary study, the angle of the shoulder joint with the most significance for the sport was considered, and all the following procedures were performed based on this angle. The sample was constituted by convenience, being formed by federated athletes who, at the time of collection, lived near Coimbra. In total, 30 federated badminton athletes were included in this study, belonging to 3 of the teams currently competing in national competitions in Portugal. The rackets were held by 15 male participants and 15 female participants, all of whom used their right hand. Mean ages were $21.00 \pm 5.60$ years, and experience in the sport was $8.40 \pm 6.93$ years. In addition, they practiced in an average of $2.93 \pm 1.26$ training sessions per week (Table 1). Subjects who presented the following conditions were excluded: (a) neurological injuries; (b) hearing or vestibular damage; (c) history of traumatic injuries to the shoulder, such as fractures, dislocations, subluxations, or surgeries; (d) any condition that causes pain or symptoms that affect active shoulder movements; (e) adverse reactions or contraindications to ice, such as areas with altered sensitivity or irritation, ice allergies, or Raynaud's disease; (f) present ranges of motion outside of normal movement patterns; (g) participating in proprioceptive recovery programs; and (h) consuming medication that could affect the conditions of the experiment [6,7,17].

**Table 1.** Sample characteristics.

| | Sample (n = 30) | | | | Male (n = 15) | | Female (n = 15) | |
|---|---|---|---|---|---|---|---|---|
| | Min | Max | Mean | SD | Mean | SD | Mean | SD |
| Age (years) | 15.00 | 34.00 | 21.00 | 5.60 | 21.47 | 5.71 | 20.53 | 5.64 |
| Weight (kg) | 47.00 | 96.40 | 64.77 | 11.33 | 72.39 | 10.16 | 57.15 | 6.18 |
| Height (cm) | 1.52 | 1.87 | 1.70 | 0.10 | 1.78 | 0.05 | 1.62 | 0.05 |
| IMC (kg/m$^2$) | 17.00 | 27.00 | 21.83 | 2.34 | 22.33 | 2.58 | 21.33 | 2.02 |
| Weekly training sessions | 1.00 | 6.00 | 2.93 | 1.26 | 3.33 | 1.35 | 2.53 | 1.06 |
| Years of badminton practice | 1.00 | 27.00 | 8.40 | 6.93 | 10.00 | 7.08 | 6.80 | 6.62 |

$n$ = sample; Min = Minimum; Max = Maximum; SD = Standard Deviation.

*2.2. Assessment*

This study assessed, using 4 time points, 2 components of proprioception: JPS and MTS. Before any assessment, all subjects were informed about the purpose, the related procedure's benefits and the risks involved in this study. Informed consent was required from all participants before participation in the study, and they were guaranteed that they could withdraw at any time without justification. The inclusion criteria for the study were as follows: Portuguese badminton federated athletes competing in the previous season and athletes aged between 15 to 35 years old. In addition, subjects were included if they had at least one year of badminton practice, and practice in, on average, at least one badminton training session per week.

### 2.2.1. Anthropometric Parameters Assessment

With the participants barefoot, anthropometric parameters such as weight (kg), using the body composition scale and height (m), using the stadiometer, were measured. This last measurement was performed on the midline of the posterior aspect of the dominant arm, midway between the shoulder and the elbow, with the upper limb relaxed. The athletes were also asked about the number of weekly badminton training sessions they practiced in, their years of badminton practice, and any history of injury in the dominant upper limb [16] (Table 1).

### 2.2.2. Isokinetic Dynamometer Setup Preparation

After the anthropometric parameters assessment, all subjects performed a JPS and MTS evaluation using an isokinetic dynamometer. Therefore, an isokinetic dynamometer Biodex System 3 (Biodex Medical Systems, New York, NY, USA) with a shoulder adapter was used to assess the proprioception data. The shoulder adapter was adjusted individually for all subjects. Before any measurement, the dynamometer's range of motion was calibrated to anatomical degrees. The isokinetic dynamometer was placed in a position that allowed the participant to perform the respective tests without constraints. The shoulder adapter was placed in a position of 90° of rotation and 30° of inclination in order to allow an adjusted standing position (Figure 1).

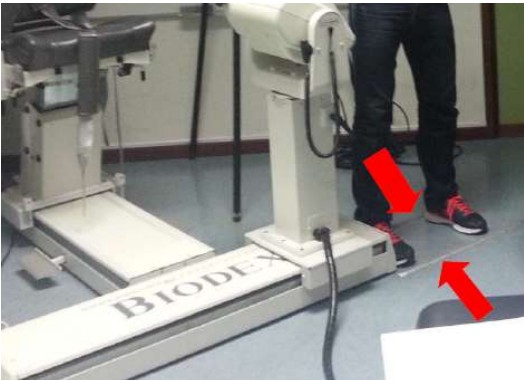

**Figure 1.** Position of the isokinetic dynamometer and its adapter used to assess the shoulder.

Each athlete was told to imagine a badminton hit above their head and after that, to state which height seemed most adjusted to them. Ten centimeters was maintained between the dynamometer and the area at which the athlete was told to place his right foot. This position was chosen because it is the most ecological for the assessment of movement in a badminton shot. Two parallel lines were placed on the floor, to signal the location at which the participants should have always had their feet throughout the tests (Figure 2). It was also recorded that the computer monitor of the dynamometer was in front of the collection location [40]. Finally, all participants had a learning moment with the isokinetic dynamometer (T0) before the main evaluation (Figure 3). The T0 moment was a learning

moment and the T1 moment was our control moment, in order to have a reference for comparison with the other moments (T2, T3, T4). Each athlete participated obligatorily in all time points (learning/control/experimental).

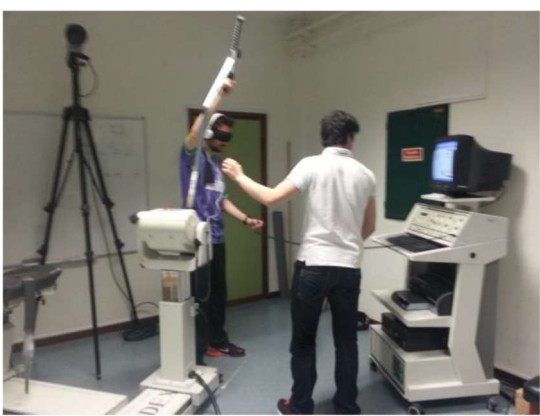

**Figure 2.** Position of the participant at the time of data collection.

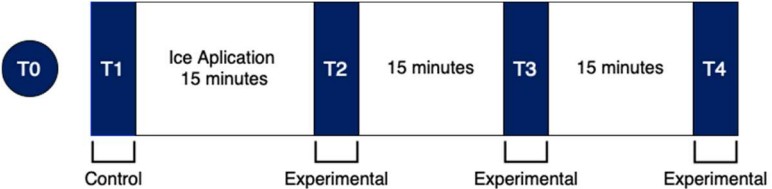

**Figure 3.** Longitudinal setup for data collection.

T-Assessment Moment

The movement that was examined using the isokinetic dynamometer was a diagonal of the shoulder, that is, a combined movement of extension, adduction, and internal rotation, to best simulate an overhead shot in badminton. All measurements took place at 149° from this shoulder diagonal (test position). This value reflects the average of the 90 shots analyzed in the pilot study. This mean value was chosen because the amplitudes presented for each shot are very similar to each other. If an individual analysis was carried out, the error associated with the measurements would be greater than the differences between the values obtained for each shot, not allowing conclusions to be drawn about how the ice would influence the proprioception of each of these technical gestures [40].

2.2.3. Maximum Strength Assessment

Then, an assessment of the maximum strength of the muscles responsible for overhead shots in badminton was performed. For this, the isometric strength assessment mode was used. As the dynamometer did not recognize any vertical force in the test position, it was decided that this measurement in the first position would be considered valid by the dynamometer when it was below 149° from the shoulder diagonal. To assess the maximum strength, the athlete was asked to perform 3 maximum repetitions, maintained for 3 s in the test position [6]. Between each repetition, 10 s of rest were allowed [16] After the test, the maximum strength value of the set of 3 repetitions was registered. Subsequently, 50% of this value was calculated [6,35] in order to have a reference for the assessment of the MTS [40].

2.2.4. Proprioception Assessment

Before measuring proprioception, the ambient temperature was checked, which should be kept constant at 20 °C [30]. All athletes were exposed to the same therapeutic conditions by maintaining the same room temperature. Then, JPS and MTS were evaluated in a random order: The components of proprioception were evaluated at 4 different time points

(Figure 3). T1 was a control procedure and served as a baseline for the experimental procedure. After this moment, ice was applied for 15 min and then 3 more measurements of proprioception were taken with a 15 min interval between them. Before this main evaluation, a learning moment (T0) with the isokinetic dynamometer was preserved [40]. There are often short intervals between games during badminton competition days, so those evaluations recreate competition aspects similar to those during competitions.

The intervention consisted of the application of 1 kg of solid ice cubes, inside a plastic bag (30 × 40 cm), for 15 min. The bag was carefully sealed and all the air in it was removed. The application was made on the anterior edge of the acromion, covering the deltoid muscle and the lateral edge of the shoulder blade. The ice bag was supported by a non-adhesive elastic bandage without compression [16]. Before the evaluation of each component of proprioception, the participants were given detailed instructions regarding the procedures that would be carried out during the data collection. This information was transmitted in a clear, objective, and identical way to all individuals, to minimize any associated experimental errors [40].

### 2.2.5. Active JPS Assessment

The dynamometer was placed in the shoulder diagonal proprioception assessment mode in the standing position. Before the test, the subject was blindfolded and used headphones playing white noise [6]. The dynamometer switch that was used to identify the positions was placed in the participant's non-dominant hand. In the first moment of the test, the athlete's dominant upper limb was taken to the starting position, that is, to the maximum possible angular position of the diagonal of the shoulder that the dynamometer allowed (about 110°). Then, passively, the dynamometer moved the anatomical segment to the test position (149° from the shoulder diagonal), giving each participant 5 s to memorize this position [6]. After that, it moved again passively to the starting position. After 10 s of rest [16], the initiation order was given to the athlete with a light tap on the non-dominant shoulder, for them to actively move their upper testing limb to the previous position, and activate the dynamometer switch when they considered that the position was reached. This procedure was repeated 3 times [1,16]. The angular velocity chosen for the movement under test was 500°/s, that is, the maximum velocity provided by the isokinetic dynamometer, as it was considered the one most adjusted to the simulation of the intended badminton shots.

### 2.2.6. MTS Assessment

The MTS measurement was very similar to the assessment of maximum strength already described, although the objective here was to reproduce only 50% of this maximum value [6,35]. The isometric strength assessment mode was used. Throughout the entire measurement, the participant's dominant upper limb was always in the position of 149° from the shoulder diagonal. Three repetitions [1] were held for 3 s [6], where the athlete tried to reproduce 50% of the previously evaluated maximum strength [6,35]. Between each repetition, 10 s of rest were provided [16]. In this first phase, the individual received visual feedback through the computer monitor of the dynamometer, so that they could comprehend the amount of force that corresponded to their 50%. In the second part of the test, the individual was blindfolded and the same procedure was repeated. During this phase, verbal instructions were given so that the participant knew when to start pushing [6].

### 2.3. Data Processing and Analysis

After evaluating the proprioception characteristics, the results obtained were recorded in degrees (JPS) and N.m (MTS). Subsequently, the absolute error, relative error, and variable error of the 3 repetitions of each measurement were calculated for the entire follow-up. The absolute error indicates the precision in terms of the global amplitude of the error, without reflecting the direction of the error, that is, if it falls below or beyond the expected value; the relative error conveys the difference between the actual angle and the target angle, considering the direction of this difference; and the variable error expresses the

variation, or consistency of the results, by providing an estimate for the precision and is obtained through the standard deviation of the different values of relative error. All error assessments were realized according to the study produced by Costello et al., (2012) [1,6]. Anthropometric characteristics were also examined.

### 2.4. Statistical Analysis

The data were statistically processed with the PASW Statistics 18 software (IBM-SPSS Statistics Corporation, New York, NY, USA). For the characterization of the sample, simple descriptive statistics were used, that is, the values of the mean, maximum, minimum, and standard deviation $(p < 0.005)$. Nominal qualitative variables (gender and dominant shoulder) and quantitative variables (age, weight, height, Body Mass Index, weekly training, and years of practice of the sport) were used in this analysis. The verification of the normality of the distribution of dependent variables was evaluated using the Shapiro–Wilk test. Then, a descriptive analysis was carried out, using the mean and standard deviation values of the dependent variables of the study, at the various measurement times. To verify the statistical significance of the differences between the two different temporal times points and within the groups, the Wilcoxon test was used for variables without normal distribution and the *t*-test was used for paired samples for variables with it. The significance level was set at $p < 0.05$.

### 3. Results

The results of the means of the absolute error, relative error, and variable error of the JPS and MTS during all assessments are depicted in Figure 4.

The results of the comparison of the different JPS and MTS errors between the time T1 and each of the other time points (T2, T3 and T4) are presented in Table 2. In addition, the only statistical significant difference was found regarding JPS when analyzing the variable error between the group. This result was found when comparing T1 and T4 $(p = 0.022)$.

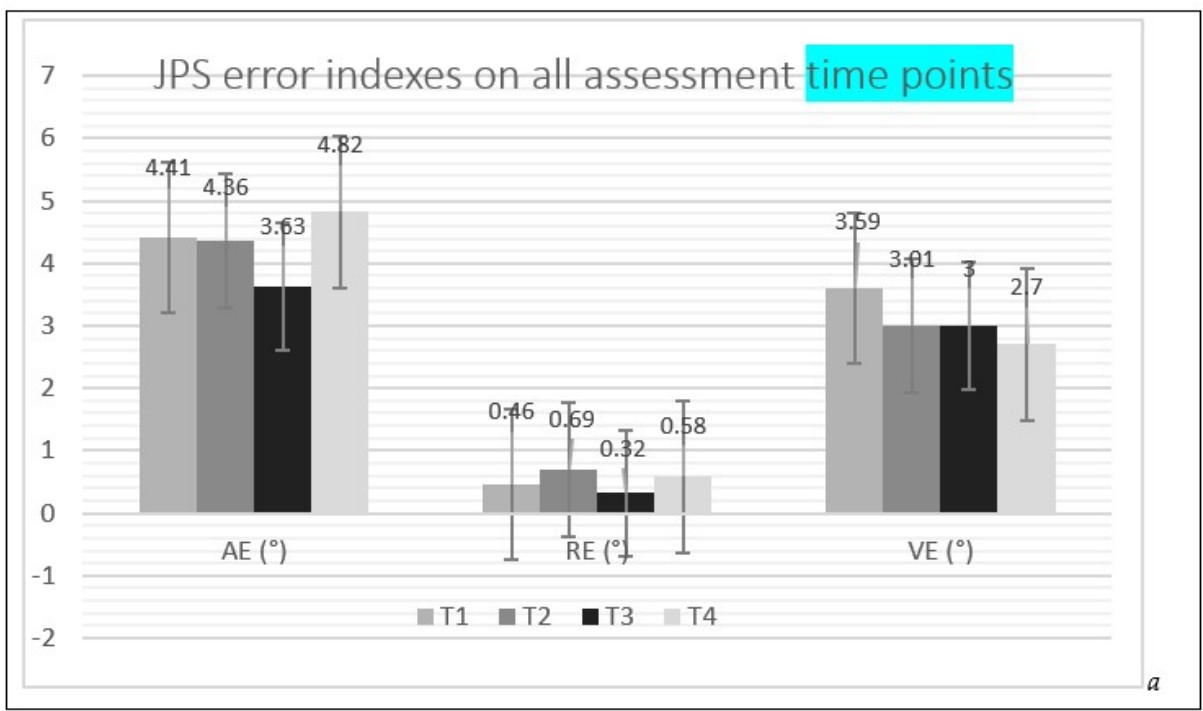

**Figure 4.** *Cont.*

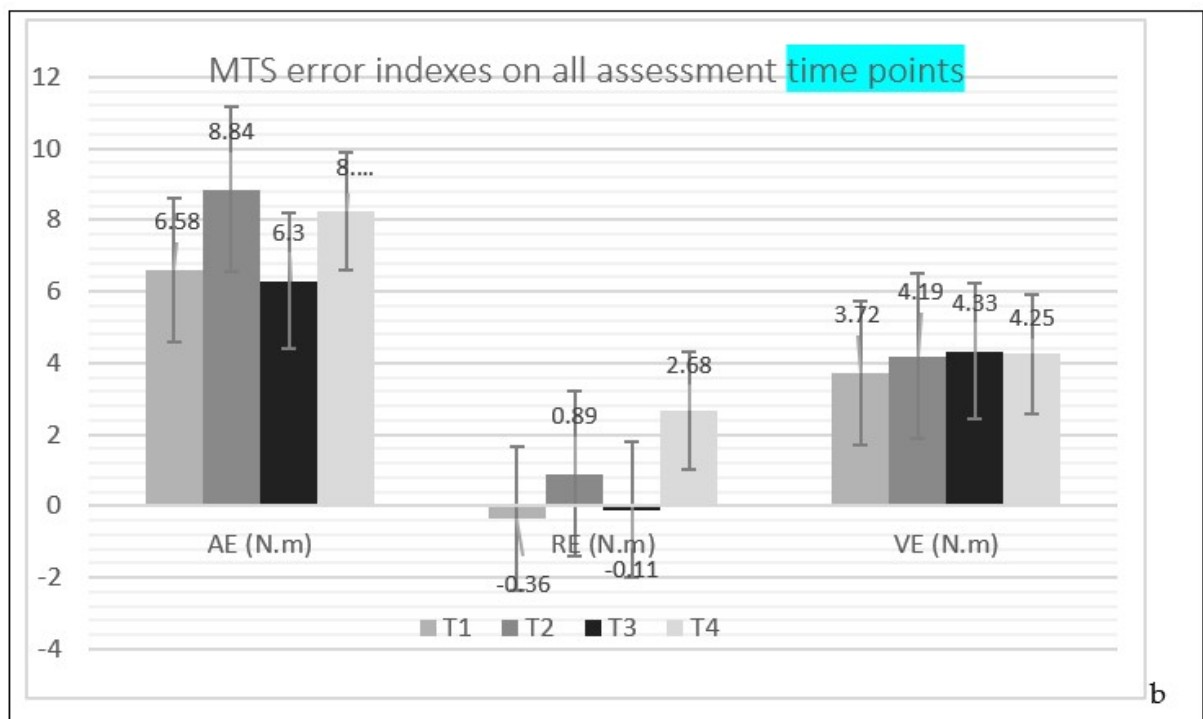

**Figure 4.** JPS (**a**) and MTS (**b**) error indexes on the four assessment time points (T1, T2, T3, T4), AE = absolute error; RE = relative error; VE = variable error.

**Table 2.** Statistical analysis regarding AE, RE, and VE of JPS and MTS, between time points T1–T2, T1–T3 and T1–T4.

| | | | Control | Experimental | Wilcoxon Test | |
|---|---|---|---|---|---|---|
| | | | | | Z | *p*-Value |
| JPS | T1–T2 Before ice application – after ice application | AE (°) | 4.41 ± 3.04 | 4.36 ± 3.33 | −0.173 | 0.863 |
| | | RE (°) | 0.46 ± 4.96 | 0.69 ± 5.16 | −0.010 | 0.992 |
| | | VE (°) | 3.59 ± 1.94 | 3.01 ± 1.70 | −1.583 | 0.113 |
| | T1–T3 15 min After ice application T1–T4 30 min After ice application | AE (°) | 4.41 ± 3.04 | 3.63 ± 2.23 | −1.310 | 0.190 |
| | | RE (°) | 0.46 ± 4.96 | 0.32 ± 3.72 | −0.134 | 0.894 |
| | | VE (°) | 3.59 ± 1.94 | 3.00 ± 1.82 | −1.321 | 0.187 |
| | | AE (°) | 4.41 ± 3.04 | 4.82 ± 3.40 | −0.878 | 0.380 |
| | | RE (°) | 0.46 ± 4.96 | 0.58 ± 5.68 | −0.260 | 0.795 |
| | | VE (°) | 3.59 ± 1.94 | 2.70 ± 1.76 | −2.283 | 0.022 |
| MTS | T1–T2 Before ice application – after ice application | AE (N.m) | 6.58 ± 6.27 | 8.84 ± 8.07 | −0.812 | 0.417 |
| | | RE (N.m) | −0.36 ± 8.89 | 0.89 ± 11.89 | −0.370 | 0.711 |
| | | VE (N.m) | 3.72 ± 2.17 | 4.19 ± 2.81 | −0.319 | 0.750 |
| | T1–T3 15 min After ice application T1–T4 30 min After ice application | AE (N.m) | 6.58 ± 6.27 | 6.30 ± 4.53 | −0.216 | 0.829 |
| | | RE (N.m) | −0.36 ± 8.89 | −0.11 ± 7.37 | −0.103 | 0.918 |
| | | VE (N.m) | 3.72 ± 2.17 | 4.33 ± 2.91 | −0.792 | 0.428 |
| | | AE (N.m) | 6.58 ± 6.27 | 8.23 ± 5.70 | −1.399 | 0.162 |
| | | RE (N.m) | −0.36 ± 8.89 | 2.68 ± 9.34 | −1.471 | 0.141 |
| | | VE (N.m) | 3.72 ± 2.17 | 4.25 ± 3.17 | −0.442 | 0.658 |

AE = absolute error; RE = relative error; VE = variable error; JPS = joint position sensation; MTS = muscle tension sensation.

## 4. Discussion

### 4.1. Descriptive Analysis

Studying the effects of cryotherapy on badminton shoulder proprioception is the main objective of the study. The learning moment T0 was used to reduce the learning bias. T1 was a control moment and T2, T3 and T4 were experimental time points (Figure 3). Differences between T1-T2, T1-T3 and T1-T4 were considered for detailed analysis (Figure 4).

### 4.2. Changes in Proprioception after Ice Application

Two of the studies on the effect of ice on shoulder JPS present opposite results to the present study [16,18] and one shows some similar results [17]. In this study, the difference between the absolute error, relative error, and variable error before and after the application of ice is also not statistically significant. Although Dover and Powers., (2004) have measured the JPS using an inclinometer, it is possible to verify that the absolute error and variable error results obtained for the internal rotation movement are rather similar to the values obtained in the present investigation. As shown in Table 2, the differences in the absolute error, relative error, and variable error of the JPS between the control moment T1 and the experimental moment T2 were not statistically significant (*p*-value > 0.05). However, such a similarity is no longer present in the results obtained by these authors for RE.

The fact that the values obtained by Dover and Powers., (2004) for the internal rotation are closer to those of the present investigation may be related to the evaluated movement [17]. The diagonal of the shoulder that was used combines the movements of the extension, adduction, and internal rotation of the shoulder, which is one of the main movements of this diagonal; this could be an explanation for the observed similarity.

On the contrary, the other two studies indicate that the changes that occurred after the application of ice were statistically significant. In the case of the study by Wassinger et al., (2007), the fact that the shoulder JPS measurement is performed using a different method, using linear displacements, makes comparison impossible [18]. However, this is no longer the case in the study by Duarte et al., (2008) study, which uses a methodology very similar to ours, although the sample is separated into males and females and only the absolute error of the shoulder joint position sensation is studied. The movement tested was the external rotation (75°), starting from an initial position of 90° of shoulder abduction and 90° of elbow flexion. In both groups, the authors observed a statistically significant increase in the absolute error after ice application. This increase was 1.6° for males and 1.2° for females [16].

The type of movement evaluated may have some influence on the JPS of the shoulder after the application of ice. When the tested movement has internal rotation, the individuals show fewer changes after the application of ice, which no longer happens when there is the external rotation of the shoulder in the evaluated movement. In addition, the type of population could influence the results. It is expected that badminton athletes present more accurate and constant values in terms of shoulder JPS, as this is one of the fundamental elements during the execution of a stroke. The fact that this component of proprioception is more developed in this population may reduce the effect of an external agent such as ice on the mechanoreceptors that are responsible for transmitting the position of the joint to the CNS. If this is the case, it is to be expected that in the study population, the JPS results are similar between T1 and T2.

The differences In the absolute error, relative error, and variable error of the MTS (Table 2), between the control moment T1 and the experimental moment T2, did not prove to be statistically significant (*p*-value > 0.05). When the MTS values are compared to the JPS values, it is possible to verify that after the application of ice, the absolute error and variable error values have different behaviors in each component of proprioception. While the mean of the absolute error and variable error in JPS decreased slightly from T1 to T2, in MTS, there was an increase in these errors after the application of the cryotherapy method (Table 2).

Given the results, we can consider that after ice is applied to the shoulder, the badminton athlete will contact the shuttlecock in the desired position, although not with the desired force. Even though these results do not show statistical significance, a difference of 2 N.m in the intended force may make some difference in a badminton stroke and be sufficient to affect the athlete's performance. In the present study, the success of the hit was not considered, but since badminton is one of the fastest sports, this aspect should be considered in future studies.

A study by Surenkok et al., (2008) analyzes the knee JPS of 15 male basketball players using the Cybex isokinetic dynamometer. Measurements were taken before and after applying a cold gel pad and spray. The JPS was studied at 45° of knee flexion, starting from two different initial positions: 0° of flexion (where the movement analyzed was flexion) and 90° of flexion (where the movement analyzed was that of extension). The absolute value of the differences in the target position after the athletes utilized the cryotherapy technique varied between 0.13° and 2.01°. All differences found were considered statistically significant by the authors [7]. These differences are very small, and are in the order of those obtained in our analysis of the MTS. Thus, although there are no statistically significant differences, the possibility that they still affect the performance of athletes is a question to be asked.

Finally, badminton is a racket sport in which there is no contact with the opponent and in which athletes are required to repeatedly perform jumps, lunges, changes in direction, and quick movements of the upper limbs in different postural positions [8]. In this sport, injuries occur mainly in the ankle, knee, and shoulder [8,9]. These are usually traumatic joint/muscular problems [10–13], or chronic injuries resulting from the overload imposed during the continuous repetition of the same sporting gestures that are characteristic of this sport [14,15]. The application of ice is a typical procedure used by physical therapists in these clinical contexts [4,16–19]. However, its use causes a decrease in the speed of the conduction of the nerve impulse in sensory afferent fibers, as well as in the excitability of muscle mechanoreceptors [4,16–18]. This physiological mechanism may alter the athlete's proprioception, leading to an increase in the risk of injury, especially in cases where the cryotherapy method is applied just before sports practice.

*4.3. Effect of Ice on Proprioception over Time*

Another issue analyzed in this investigation is the behavior of proprioception over time, after an application of ice. Regarding joint position sensation, there is a decrease between 0 and 15 min, and an increase from 15 to 30 min, both in the absolute error and relative error. In turn, variable error tends to decrease progressively over 30 min (Table 2). It is also noted that the differences between the measurement time points were not statistically significant, except for the joint position variable error between T1 and T4, which presented a *p*-value of 0.022 (Table 2).

Regarding the assessment of the joint position sensation, it appears that in the first 15 min, the absolute error and relative error decrease, however, from 15 to 30 min, they tend to increase. The variable error, on the other hand, has a very small variation (about 0.14 N.m) over the 30 min: in the first 15 min, it decreased and then increased until 30 min (Table 2). None of the differences between the various time points proved to be statistically significant (Table 2).

None of the studies that identified the effect of ice on proprioception analyzed the time after the application of the cryotherapy technique. It is only possible to find this temporal analysis in investigations that study JPS in the knee and ankle joints. Regarding the knee joint, five articles were found to perform this analysis. In four of these investigations, the evaluation of the moment over time is performed only once, after 15/20 min [19,22,33,34]. The other study performs measurements after 5, 10, 15, 20, 25, and 30 min of cryotherapy application [32]. In three of these investigations [32–34], the authors concluded that the application of ice significantly interferes with the individuals' proprioception. Regarding the ankle joint, two studies were found that make a temporal analysis of the effect of

ice [24,30]. In both cases, JPS measurements were taken 15 min after ice application. The differences were considered statistically significant in only one of the situations [30].

It is therefore possible to verify that there is no consensus on the effect of ice over time on proprioception. In the present study, only the decrease in the variable error of JPS between T1 and T4 was statistically significant; however, given the particularity of this result, this may have been caused by the learning bias inherent in the experience. Although most of the differences obtained were not considered statistically significant, it should be noted that in both components of proprioception, there was an increase in the absolute error and relative error between moment T3 (15 min) and moment T4 (30 min) (Table 2). The main explanatory reason for this decrease in proprioception is related to the athletes' possible tiredness at the last measurement moment. This fatigue may be physical, due to the muscular force exerted in high shoulder amplitudes, as well as psychological, due to the total duration of the experimental activity (about 75 min).

### 4.4. Study Limitations

In the present study, the first training moment measurement that was made with the dynamometer was performed on the same day as the data collection, T0 time points, although written and verbal information about the experimental procedure had been previously communicated. For that reason, there is some repetition bias associated with the number of assessments performed by each athlete. It is normal that, throughout the data collection process, the participant was more prepared and trained to perform the requested tests, compared to the first measurements and the initial T0 time point. Although every precaution was taken to try to best simulate a badminton hit above the head, it is important to note that there are some differences between the real movement and the evaluated diagonal. The exact reproduction of the movement to be studied would only be possible in the field and not in the laboratory, which is the context of this research.

Lastly, it is important to note that the present study examined healthy athletes without shoulder problems to see if cryotherapy diminishes proprioception and adversely affects performance. However, in a practical context, the athlete uses cryotherapy when a change in the shoulder causes pain. That is, in a real context, the proprioception of athletes with shoulder pain would be different from what was measured in this sample of thirty athletes.

### 5. Conclusions

Thus, given the results obtained, we can conclude that after a 15 min application of 1 kg of broken solid ice, badminton athletes will be able to return immediately to the practice of the sport, as there does not seem to be an increased risk of injury associated with a deficit in proprioception and thus an impairment in their performance in subsequent games. Therefore, regarding these conditions, ice can be considered a safe therapeutic intervention, as it also promotes an analgesic effect at the site of injury. However, it is important to clarify other aspects in future investigations, such as the success of shootings after the application of cryotherapy.

**Author Contributions:** Conceptualization, N.T. and M.A.C.; methodology, N.T. and M.A.C.; software, N.T. and M.A.C.; validation, N.T. and M.A.C.; formal analysis, N.T. and M.A.C.; investigation, N.T. and M.A.C.; resources, M.A.C. and. G.D.; data curation, N.T. and M.A.C.; writing—original draft preparation, N.T. and J.M.; writing—review and editing, M.A.C., J.M. and G.D.; visualization, N.T. and J.M.; supervision, M.A.C. and G.D. All authors have read and agreed to the published version of the manuscript.

**Funding:** This research received no external funding.

**Institutional Review Board Statement:** The study was conducted according to the guidelines of the Declaration of Helsinki and approved by the Ethics Committee of Polytechnic Institute of Coimbra (CEPC 7/2018).

**Informed Consent Statement:** Informed consent was obtained from all subjects involved in the study.

**Data Availability Statement:** Not applicable.

**Acknowledgments:** M.A.C. and J.M. acknowledge the support of the Centre for Mechanical Engineering, Materials and Processes—CEMMPRE of the University of Coimbra, which is sponsored by Fundação para a Ciência e Tecnologia (FCT) (UIDB/00285/2020, LA/P/0112/2020). The authors acknowledge ROBOCORP, i2A, co-funded by QREN under the Programa Mais Centro of the Coordination Commission of the Central Region and the European Union through the European Regional Development Fund.

**Conflicts of Interest:** The authors declare no conflict of interest.

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
