# Peer review of "The Effect of Ice on Shoulder Proprioception in Badminton Athletes"

_ejihpe, doi:10.3390/ejihpe13030051_

Round 1

Reviewer 1 Report

Congratulations on the manuscript entitled "Ice Effect on Shoulder Proprioception in Badminton Athletes" which shows the importance of applying ice in different time frames in badminton players. However, it is considered to take into account some considerations so that it can be published in this journal:

Introduction:

It is complete and presents and justifies the selected problem in a deep and adequate way.

Materials and Methods:

The methodology carried out is adequately and precisely shown to be replicated if necessary.

Results:

It is recommended to briefly expand this section. It is recommended to briefly present the results, highlighting those that are considered most important due to their relevance.

Discussion:

It is recommended to start this section by exposing the objective(s) of the study and then highlight the most relevant aspects obtained from it, and then move on to the discussion with the other studies. In addition, although the subsections are appropriate, it is recommended to structure them in a different way, comparing the results obtained in this study with the others, and giving a possible reason why they differ or are similar to each other. On the other hand, the relationship between the results obtained and the injuries is not sufficiently justified. This needs to be clarified and further developed (Influencing the final conclusions of the study).

With these recommendations, it is considered that the quality of the manuscript may be optimal for its subsequent publication.

Thank you very much and greetings.

Author Response

Response to the reviewer’s comments:

The authors would like to thank the reviewers for reading the manuscript and for the comments and suggestions to improve the manuscript. Changes in the revised manuscript are highlighted in blue color.

Response to Reviewer 1 Comments

Reviewer #1: Congratulations on the manuscript entitled "Ice Effect on Shoulder Proprioception in Badminton Athletes" which shows the importance of applying ice in different time frames in badminton players. However, it is considered to take into account some considerations so that it can be published in this journal: 

Point 1 : Introduction: It is complete and presents and justifies the selected problem in a deep and adequate way.

Response 1: n/a

Point 2 : Materials and Methods: The methodology carried out is adequately and precisely shown to be replicated if necessary.

Response 1: n/a

Point 3: Results: It is recommended to briefly expand this section. It is recommended to briefly present the results, highlighting those that are considered most important due to their relevance.

Response 3: As recommended by the reviewer, we realizes alterations according to the suggestions. The text can be now read as:

“The results of the comparison of the different JPS and MTS errors between the time T1 and each of the other time points (T2, T3 and T4) are presented in Table 2. Also, the only statistical significant difference were found regarding JPS when analyzing Varibale Error between group. This result were found when comparing T1 and T4 (p=0.022).”

Point 4: Discussion: It is recommended to start this section by exposing the objective(s) of the study and then highlight the most relevant aspects obtained from it, and then move on to the discussion with the other studies. In addition, although the subsections are appropriate, it is recommended to structure them in a different way, comparing the results obtained in this study with the others, and giving a possible reason why they differ or are similar to each other. On the other hand, the relationship between the results obtained and the injuries is not sufficiently justified. This needs to be clarified and further developed (Influencing the final conclusions of the study).

Response 4: As recommended by the reviewer, we realizes alterations according to the suggestions. The text can be now read as:

Studying the effects of cryotherapy on badminton shoulder proprioception is the main objective of the study. The learning moment T0 was used to reduce the learning bias. T1 was a control moment and T2, T3 and T4 were a experimental time points (Figure 3). Differences between T1-T2, T1-T3 and T1-T4 were considered for detailed analysis (Figure 4)….

….Finally, badminton is a racket sport where there is no contact with the opponent and which requires athletes to repeatedly perform jumps, lunges, changes of direction, and quick movements of the upper limbs in different postural positions [9]. In this modality, injuries occur mainly in the ankle, knee, and shoulder [9,10]. These are usually traumatic joint/muscular problems [11–14], or chronic injuries resulting from the overload imposed during the continuous repetition of the same sporting gestures characteristic of this modality [15,16]. The application of ice is a typical procedure used by the physical therapist in these clinical contexts [4,8,17–19]. However, its use causes a decrease in the speed of conduction of the nerve impulse of sensory afferent fibers, as well as in the excitability of muscle mechanoreceptors [4,8,17,18]. This physiological mechanism may alter the athlete's proprioception, leading to an increase in the risk of injury, especially in cases where the cryotherapy method is applied just before sports practice….

…In the present study, the first training moment of the measurement with the dynamometer was performed on the same day of the data collection, T0 time points, although previous written and verbal information about the experimental procedure was done.  For that reason, there is some repetition bias associated with the number of assessments that were performed by each athlete. It is normal that, as the data collection was done, the participant was more prepared and trained to perform the requested tests, compared to the first measurements and the initial T0 time point….

…Lastly, it is important to note that the present study examined healthy athletes without shoulder problems to see if cryotherapy diminishes proprioception and adversely affects performance. However, in a practical context, the athlete uses cryotherapy when a change in the shoulder causes pain. That is, in a real context, the proprioception of athletes with shoulder pain will be different from what was measured in this sample of 30 athletes.”

Reviewer 2 Report

I would like to see an article that would bring a significant improvement to the medical world and the group of orthopedic doctors. At the moment, given the much too small group of patients, the much too usual procedure presented, I consider that it does not meet the quality criteria for the magazine to which it was applied.

Author Response

Response to Reviewer 2 Comments

Response to the reviewer’s comments:

The authors would like to thank the reviewers for reading the manuscript and for the comments and suggestions to improve the manuscript.

Reviewer #2: I would like to see an article that would bring a significant improvement to the medical world and the group of orthopedic doctors. At the moment, given the much too small group of patients, the much too usual procedure presented, I consider that it does not meet the quality criteria for the magazine to which it was applied.

Response: As our sample was carried out for convenience and which includes a very specific population, namely badminton athletes, it was difficult for us to find a sizable sample. Moreover, badminton is not a popular sport in Portugal, which means that there is even more shortage of federated athletes who can participate in our study.

Reviewer 3 Report

The authors have presented a study that has been well thought out and seem to have made a genuine effort to ensure adequate rigor, so well done.

However, I feel the manuscript needs some substantial improvement in its contextualization to strengthen the study's rationale, as well as to more fully discuss the results, not only in the context of other research, but also in relation to the practical implications of such research. This will enable a clearer sense of contribution to be made by the reader.

Some comprehensive comments have been provided below to hopefully assist the authors in strengthening the manuscript:

1. Introduction:

Re-focus the introduction to strengthen the need for the research. This should include reducing paragraph one.

Key considerations: more fully explain the uses of ice, especially in the context of the stages of healing. The manuscript alludes to the use of ice before sport, but when/why does this happen? Ice is generally applied in the acute phase after injury. Therefore, why is there concern that ice may impact proprioception, wouldn't injury impact this more than any beneficial healing effect caused by ice? Also, some attention should be paid to current recommendations/opinions of the use of cryotherapy in acute treatment of injuries being that ice application is not always seen as necessary (e.g. PEACE and LOVE methods of acute treatment), except to decrease secondary hypoxic injury. Perhaps some discussion around injuries requiring early movement to enhance tissue remodeling would be a good approach and relate this to any potential affect that ice may have on the need for proprioception during this movement.

Apart from there not being many studies, what's the rationale requiring your study of the shoulder on badminton players? Why is it not adequate to study a non-sporting sample on the effect of ice on proprioception and liken this to a sporting population - soft tissue injury is soft tissue injury, treatment protocols are similar despite the group? Highlight/explain this more fully.

One of the study's objectives is to verify how long it takes for shoulder proprioception to return after ice, however, this assumes there will be an impact - suggest rewording this aim.

2. Materials and methods

Overall, this section is comprehensive, however a clearer explanation of the testing of participants is required. For example, the authors mention a control 'procedure', does this mean that all participants completed the same number of testing protocols (randomly allocated) and acted as their own control in T0, or did you have a group that didn't have ice application? Please clarify.

Skin folds were taken of the triceps. Was there a reason biceps SF wasn't taken given ice was applied to the anterior of the shoulder. Also, the manuscript mentions the collection of surface temp during ice application, how was this monitored/checked and some explanation should be provided as to how the study accounted for variations in tissue density and cooling effect across subjects. Furthermore, what was the rationale behind the various testing time points? Given one of the study objectives was to verify how long until proprioception was returned, was 30 mins enough for this to happen, or should the testing times have been individualized depending on cooling effect reached and tissue density? Expand.

- Consider adding the participant inclusion and exclusion criteria info into section 2.1. 

- Please reword text in line 123 relating to participants' sex.

- In section 2.2, try to clearly explain (overall) what all participants were required to undertake. Perhaps consider introducing Fig 3 in this section, as T0 is referred to here, but is out of context, as Fig 3 has not been specifically mentioned. Also, more concisely explain the ethical procedures - too text heavy.

- When referring to 'moments', consider changing this to "time points" throughout manuscript. Also, 'learning moment' (line 135) should be referred to as a "familiarization". Change throughout manuscript.

- Line 170 and 172 - change 'place' to "location". Check throughout manuscript.

- Line 171 - change 'preserved' to "recorded". Check throughout manuscript.

- Line 199 - was room temp kept constant or just checked? - clarify

- Line 205 - correct spelling error and change 'preserved' to "recorded".

- In section 2.2.5, can the authors more clearly explain the dynamometer protocol? What is meant by 250 degrees at the shoulder diagonal - seems anatomically impossible.

- Line 238 - remove 'already' and change to "described in section 2.2.3" and change 'is' to "was".

- Info in lines 259-60 needs expansion (see earlier comment).

- Section 2.4 - why the need for italic text? Possibly check this throughout manuscript. Check line 273 - seems to have an error (not a sentence).

4. Results

- Figure 4 should be presented as Fig 4a and 4b I would have thought.

- In Tabe 2, move "MTS" to top of column (not center).

5. Discussion

Overall, this section needs a lot more work.

The opening statement does not address the strong possibility that a learning effect was probably a key contributor in the study's overall findings.

The authors need to again pick up on the point of why it is important to study the impact of ice on proprioception here - why is ice the issue to proprioception and not the injury itself? This is probably a key point in the paper and is unfortunately included as a limitation at the very end of the paper. Hence, why a reconfiguration of the intro is required to perhaps focus on early movement for remodeling/healing of tissue and the possible impact that ice may have on proprioception for this.

In a practical sense, in the overall context of soft tissue injury management and healing, wouldn't practitioners want the effect of ice to last and not subside quickly? Again, why is a loss of proprioception an issue, which this study shows it isn't? Perhaps add a 'practical implications' section.

- Line 298 - fix reference error

- Line 361 - add acronym

- Line 418 - define 'shootings'

Author Response

Response to Reviewer 3 Comments

Response to the reviewer’s comments:

The authors would like to thank the reviewers for reading the manuscript and for the comments and suggestions to improve the manuscript.

Reviewer #3: The authors have presented a study that has been well thought out and seem to have made a genuine effort to ensure adequate rigor, so well done. However, I feel the manuscript needs some substantial improvement in its contextualization to strengthen the study's rationale, as well as to more fully discuss the results, not only in the context of other research, but also in relation to the practical implications of such research. This will enable a clearer sense of contribution to be made by the reader. Some comprehensive comments have been provided below to hopefully assist the authors in strengthening the manuscript:

  1. Introduction: Introduction: Re-focus the introduction to strengthen the need for the research. This should include reducing paragraph one. Key considerations: more fully explain the uses of ice, especially in the context of the stages of healing. The manuscript alludes to the use of ice before sport, but when/why does this happen? Ice is generally applied in the acute phase after injury. Therefore, why is there concern that ice may impact proprioception, wouldn't injury impact this more than any beneficial healing effect caused by ice? Also, some attention should be paid to current recommendations/opinions of the use of cryotherapy in acute treatment of injuries being that ice application is not always seen as necessary (e.g. PEACE and LOVE methods of acute treatment), except to decrease secondary hypoxic injury. Perhaps some discussion around injuries requiring early movement to enhance tissue remodeling would be a good approach and relate this to any potential affect that ice may have on the need for proprioception during this movement. Apart from there not being many studies, what's the rationale requiring your study of the shoulder on badminton players? Why is it not adequate to study a non-sporting sample on the effect of ice on proprioception and liken this to a sporting population - soft tissue injury is soft tissue injury, treatment protocols are similar despite the group? Highlight/explain this more fully.

Response: As recommended by the reviewer, we realizes alterations according to the suggestions. The text can be now read as:

“Proprioception is based on the transmission to the Central Nervous System (CNS) of afferent information from muscle, tendon, joint capsule, ligament, and skin [1]. For a certain motor action to be performed successfully, the afferent information must be correctly transmitted to the CNS, through the visual, vestibular, and somatosensory sensory systems. processing of this afferent information, performed at the central level, will later be used to generate an efferent response that allows the proper regulation of muscle tone, ensuring correct patterns of joint stability, coordination, and balance during human movement [2]. Analogously, it is possible to compare the peripheral mechanoreceptors to the hardware of a computer that provides proprioceptive information for the software (central processing) to integrate and use [3]. Through this process, the individual can compare the real movement with the intended movement and thus develop his motor learning [1]. That is, learning motor skills means developing new movement patterns by processing appropriate proprioceptive information [3]….

…The application of different cryotherapy methods is frequently used in the clinical practice of physiotherapy in a sports environment, as its analgesic and anti-inflammatory properties have been associated with a faster recovery from injuries to the neuro-musculoskeletal system [4,8,17–19]. Cryotherapy serves as a common treatment method for acute sports injuries. Analgesia, inflammation reduction and secondary hypoxic injuries are commonly treated with this method. When an athlete returns to the sport immediately after a serious injury, the benefits of cryotherapy might be questioned since suppressing pain sensations could lead to more tissue damage. In addition, a study suggests that cryotherapy does not reduce delayed-onset muscle soreness following endurance and strength training. [20]

  • One of the study's objectives is to verify how long it takes for shoulder proprioception to return after ice, however, this assumes there will be an impact - suggest rewording this aim.
  • Response: Altered in the text. The text can be now read as:

…The main objective of this study is to analyze the influence of the application of a cryotherapy method on shoulder proprioception in badminton athletes.”

  1. Materials and methods: Overall, this section is comprehensive, however a clearer explanation of the testing of participants is required. For example, the authors mention a control 'procedure', does this mean that all participants completed the same number of testing protocols (randomly allocated) and acted as their own control in T0, or did you have a group that didn't have ice application? Please clarify. Skin folds were taken of the triceps. Was there a reason biceps SF wasn't taken given ice was applied to the anterior of the shoulder. Also, the manuscript mentions the collection of surface temp during ice application, how was this monitored/checked and some explanation should be provided as to how the study accounted for variations in tissue density and cooling effect across subjects. Furthermore, what was the rationale behind the various testing time points? Given one of the study objectives was to verify how long until proprioception was returned, was 30 mins enough for this to happen, or should the testing times have been individualized depending on cooling effect reached and tissue density? Expand.

Response: As recommended by the reviewer, we realizes alterations according to the suggestions. The text can be now read as:

“The rackets were held by 15 male participants and 15 female participants, all of whom used their right hand. Mean ages were 21.00 ± 5.60 years, and experience in the modality was 8.40 ± 6.93 years. In addition, they performed an average of 2.93 ± 1.26 training sessions per week (Table 1). Subjects who presented the following conditions were excluded: a) neurological injuries; b) hearing or vestibular damage; c) history of traumatic injuries to the shoulder such as fractures, dislocations, subluxations, or surgeries; d) any condition that causes pain or symptoms that affect active shoulder movements; e) adverse reactions or contraindications to ice, such as areas with altered sensitivity or irrigation, ice allergies, or Raynaud's disease; f) present ranges of motion outside of normal movement patterns; i) participating in proprioceptive recovery programs; j) consuming medication that could affect the conditions of the experiment [6,7,17]….

…This study assessed on four time points 2 components of proprioception: JPS and MTS. Before any assessment, all subjects were informed about the purpose, the related procedure’s benefits and risks involved in this study. Informed consent was required from all participants before participation in the study, and they were guaranteed that they could withdraw at any time without justification….

…Finaly, all participants had a learning moment with the isokinetic dynamometer (T0) before the main evaluation (Figure 3). The T0 moment was a learning moment and the T1 moment was our control moment, in order to have a reference for comparison with the other moments (T2, T3, T4). Each athlete participated obligatorily in all times points (learning / control / experimental)….

…There are often short intervals between games during badminton competition days, so those evaluations recreate competition aspects similar to those during competitions.”

  • Consider adding the participant inclusion and exclusion criteria info into section 2.1. 
  • Response: Altered in the text.

  • Please reword text in line 123 relating to participants' sex.
  • Response: Altered in the text.

  • In section 2.2, try to clearly explain (overall) what all participants were required to undertake. Perhaps consider introducing Fig 3 in this section, as T0 is referred to here, but is out of context, as Fig 3 has not been specifically mentioned. Also, more concisely explain the ethical procedures - too text heavy.
  • Response: Altered in the text.

  • When referring to 'moments', consider changing this to "time points" throughout manuscript. Also, 'learning moment' (line 135) should be referred to as a "familiarization". Change throughout manuscript.
  • Response: Altered in the text.

  • Line 170 and 172 - change 'place' to "location". Check throughout manuscript.
  • Response: Altered in the text.

  • Line 171 - change 'preserved' to "recorded". Check throughout manuscript.
  • Response: Altered in the text.

  • Line 199 - was room temp kept constant or just checked? – clarify
  • Response: Altered in the text.

  • Line 205 - correct spelling error and change 'preserved' to "recorded".
  • Response: Altered in the text.

  • In section 2.2.5, can the authors more clearly explain the dynamometer protocol? What is meant by 250 degrees at the shoulder diagonal - seems anatomically impossible.
  • Response: Altered in the text.

  • Line 238 - remove 'already' and change to "described in section 2.2.3" and change 'is' to "was".
  • Altered in the text.

  • Info in lines 259-60 needs expansion (see earlier comment).
  • Altered in the text.

  • Section 2.4 - why the need for italic text? Possibly check this throughout manuscript. Check line 273 - seems to have an error (not a sentence).
  • Altered in the text.

  1. Results

  • Figure 4 should be presented as Fig 4a and 4b I would have thought.
  • Altered in the text.

  • In Tabe 2, move "MTS" to top of column (not center).
  • Altered in the text.

  1. Discussion: Overall, this section needs a lot more work. The opening statement does not address the strong possibility that a learning effect was probably a key contributor in the study's overall findings. The authors need to again pick up on the point of why it is important to study the impact of ice on proprioception here - why is ice the issue to proprioception and not the injury itself? This is probably a key point in the paper and is unfortunately included as a limitation at the very end of the paper. Hence, why a reconfiguration of the intro is required to perhaps focus on early movement for remodeling/healing of tissue and the possible impact that ice may have on proprioception for this. In a practical sense, in the overall context of soft tissue injury management and healing, wouldn't practitioners want the effect of ice to last and not subside quickly? Again, why is a loss of proprioception an issue, which this study shows it isn't? Perhaps add a 'practical implications' section.

Response: As recommended by the reviewer, we realizes alterations according to the suggestions. The text can be now read as:

“Studying the effects of cryotherapy on badminton shoulder proprioception is the main objective of the study. The learning moment T0 was used to reduce the learning bias. T1 was a control moment and T2, T3 and T4 were a experimental time points (Figure 3). Differences between T1-T2, T1-T3 and T1-T4 were considered for detailed analysis (Figure 4)….

…Finally, badminton is a racket sport where there is no contact with the opponent and which requires athletes to repeatedly perform jumps, lunges, changes of direction, and quick movements of the upper limbs in different postural positions [9]. In this modality, injuries occur mainly in the ankle, knee, and shoulder [9,10]. These are usually traumatic joint/muscular problems [11–14], or chronic injuries resulting from the overload imposed during the continuous repetition of the same sporting gestures characteristic of this modality [15,16]. The application of ice is a typical procedure used by the physical therapist in these clinical contexts [4,8,17–19]. However, its use causes a decrease in the speed of conduction of the nerve impulse of sensory afferent fibers, as well as in the excitability of muscle mechanoreceptors [4,8,17,18]. This physiological mechanism may alter the athlete's proprioception, leading to an increase in the risk of injury, especially in cases where the cryotherapy method is applied just before sports practice….

…In the present study, the first training moment of the measurement with the dynamometer was performed on the same day of the data collection, T0 time points, although previous written and verbal information about the experimental procedure was done.  For that reason, there is some repetition bias associated with the number of assessments that were performed by each athlete. It is normal that, as the data collection was done, the participant was more prepared and trained to perform the requested tests, compared to the first measurements and the initial T0 time point…..

…Lastly, it is important to note that the present study examined healthy athletes without shoulder problems to see if cryotherapy diminishes proprioception and adversely affects performance. However, in a practical context, the athlete uses cryotherapy when a change in the shoulder causes pain. That is, in a real context, the proprioception of athletes with shoulder pain will be different from what was measured in this sample of 30 athletes….

….as there does not seem to be an increased risk of injury associated with a deficit in proprioception and thus impair their performance in subsequent games. Therefore, regarding those condition, ice can be considered a safe therapeutic intervention, as I promotes also an analgesic effect at the site injury.”

  • Line 298 - fix reference error
  • Altered in the text.

  • Line 361 - add acronym
  • Altered in the text.

  • Line 418 - define 'shootings'
  • Altered in the text.

Round 2

Reviewer 1 Report

Thank you very much for considering my recommendations

Reviewer 2 Report

I can see a lot of good changes, I’m considering now that the article it is a suitable one for the journal.

Reviewer 3 Report

Manuscript has been improved, well done. There are two spelling errors in the revised version:

- Line 172 - 'finaly' to "finally"

- Line 446 - 'I' should be changed to "ice"